# Wear Measurements in Cylindrical Telescopic Crowns Using an Active Piezoresistive Cantilever with an Integrated Gold Microsphere Probe

**DOI:** 10.3390/ma18194624

**Published:** 2025-10-07

**Authors:** Tomasz Dąbrowa, Dominik Badura, Bartosz Pruchnik, Władysław Kopczyński, Ivo W. Rangelow, Edward Kijak, Teodor Gotszalk

**Affiliations:** 1Department of Prosthodontics, Wrocław Medical University, ul. Krakowska 26, 50425 Wrocław, Poland; tomasz.dabrowa@umw.edu.pl (T.D.); edward.kijak@umw.edu.pl (E.K.); 2Department of Nanometrology, Wrocław University of Science and Technology, ul. Janiszewskiego 11/17, 50372 Wroclaw, Poland; dominik.badura@pwr.edu.pl (D.B.); wladyslaw.kopczynski@pwr.edu.pl (W.K.); teodor.gotszalk@pwr.edu.pl (T.G.); 3Department of Micro- and Nanoelectronical Systems, Technische Universität Ilmenau, Ehrenbergstraße 29, 98693 Ilmenau, Germany; ivo.rangelow@tu-ilmenau.de

**Keywords:** AFM, wear, nanometrology, biomaterials, HS-AFM, telescopic crown

## Abstract

In this paper, we report a novel application of atomic force microscopy (AFM) for measurement of wear of prosthetic materials. In contrast to previously employed methods, we introduce AFM-based wear induction. In this way, we utilize AFM as both measurement technique and the mean for surface wear. We describe the methodology along with the metrological advantages of the approach regarding the supreme resolution of volume measurement (down to 1 μm^3^). We investigate wear between prosthetic gold alloy (Degulor M) and FGP polymeric material from Bredent in nanoscale. For that purpose, we modify active piezoresistive cantilever, replacing the original tip with Degulor M microsphere. We elaborate on the process of modification and present how the mass volume and topology of the tip is controlled throughout the process. Wear process was performed in reciprocal motion over the length of 5 μm in 35,000 repetitions to mimic the actual conditions occurring in human mouth cavity. We present how this method, by focusing on a small area of investigated materials, leads to shortening the overall time of wear measurements from tong term observations down to several minutes. AFM-measured data present consistent relation between wear energy and wear volume. Exemplary results seem to confirm durability of the FGP-Degulor M mechanical contact and occurring strengthening of the mechanical contact with roughening of the polymeric surface.

## 1. Introduction

Overdentures are prosthetic restorations that cover the root surfaces of residual dentitions, pillars or intraosseous implants. Clinical studies have shown that the roots of residual dentitions protect the alveolar process from atrophy and that denture retention can be achieved through the placement of prosthetic abutments.

Telescopic crowns, which usually consist of two components, can aid in denture retention. The two components of telescopic crowns are the primary crown, which is permanently cemented to the ground pillar tooth, and a secondary crown, which closely matches the primary crown and is connected to the removable partial denture (RPD) structure. These restorations are used in cases of residual dentition due to the enhanced intermediate splinting of the pillar teeth and good retention. The retention force of a telescopic denture anchored with a cylindrical telescopic crown is mainly determined by the presence of static friction force. The force value can be influenced by the type of biomaterial used for the abutment and the accuracy of its fit. New material associations are possible with the increasing number of new prosthetic products from which telescopic crowns are made—which is the subject of ongoing investigations [1,2]. In telescopic technology, there is no strict protocol for selecting specific materials. However, investigating macroscopic prosthetics—either in vitro or in vivo—is a more expensive and time-consuming process [3,4]. Therefore, the methods for analyzing micro- and nanoscale structures should be and are developed, including correlative and combinatoric approaches [5,6,7].

As telescopic surfaces undergo wear during use, the fit loosens, causing a reduction in the retention force of the abutment [8]. We believe that the entire process is the result of nanoscale phenomena. In particular, the characteristics of molecular adhesion between the molecules covering the prosthetic crowns depend on the layer chemistry and the solvent that promotes molecular binding. Therefore, this assumption clearly indicates that a detailed analysis of surface behaviour is needed to understand and optimize restoration fitting.

In one of our previous papers [9], we presented a study based on tribological tests performed on selected biomaterials that are used in telescopic dentures. In this study, we used gold-based dental alloys, a chromium–cobalt alloy and zirconium oxide. The selected biomaterials function with a counter specimen composed of an FGP. Tribological tests were conducted in an artificial saliva environment. The tests confirmed the dependence of the static friction coefficient on the contact pressure for the analyzed pairs of materials used in dental prosthetics. The greatest influence of contact pressure on the coefficient of friction occurred in the case of friction between the zirconia ceramic and FGP composite. The most stable coefficient of friction under changing contact pressure and service life conditions was found for the Brealloy 270 cobalt alloy. In this context, atomic force microscopy (AFM) is the best method for imaging surface properties at the nanoscale. In contact AFM (C-AFM) technology, the probe contacts the investigated sample. The interaction is defined precisely by monitoring the cantilever deflection and thus can be set in the range from nanonewtons (where no tip and/or surface damage is possible) to micronewtons (where indentation is induced). Therefore, the same nanoprobe can be used as a nanoindenter or as a tool for surface imaging. Moreover, dedicated modelling of probe properties (e.g., by geometric modification or selection of mechanical or chemical composition) makes it possible to simulate interactions occurring in real systems. In this manner, mechanical phenomena, such as wear and scratching, are studied at the nanoscale for the load force between the nanoprobe and the sample, which varies in the range of up to several micronewtons.

Wear is defined as the removal of material from solid surfaces in solid-state contact [10]. Wear, defined as the energy required to remove a unit of volume, is a quantitative measure of contact behaviour. Scratching, in turn, is a process that leads to surface removal under defined conditions and may be a wear test technique. Wear can be examined equally at the nanoscale as in macroscopic methods [11]. However, the experiments conducted thus far have enabled only qualitative descriptions of phenomena [12,13]. However, metrological quantities may be applied for nanoscratching [14,15]. Conversely, the nanoscratching process can be described in terms of dynamic nanoindenter ploughing with the tip of an atomic force microscope [16].

The application of atomic force microscopes in wear or nanoscratching investigations stems from the fact that the probe replaces two separate tools—the indenter and surface scanner [17].

The application of an atomic force microscope operating in dual mode for quantitative wear analysis has been described in the literature [18]. Similar tests involving diamond probes were presented previously [19]. The pin-on-disc macroscopic approach (involving circular motion of the indenter over the sample) was used in AFM [20]. In contrast to similar tests [21], we postulate that linear scratching guarantees a more predictable wear process than areal material removal does because of the uniaxiality of the wear.

The novelty of this paper lies in the application of active piezoresistive cantilevers [22] fabricated via microelectronic technology. They integrate a piezoresistive deflection detector for tip displacement sensing, an actuator for inducing cantilever movement and a silicon probe tip with a standard radius of tens of nanometres. The resistance of the piezoresistive detector changes when the beam is deflected. The response of the detector is electrical and is measured by the external electronics. No optical setup is needed to detect beam movement, which makes it possible to integrate the cantilevers with optical or electron microscopes. The deflection actuator can operate thermomechanically or electromagnetically [23]. The active piezoresistive cantilevers can be precisely calibrated; thus, the transfer functions of the detector and actuator can be described quantitatively, independent of the system in which they are used [24], and cannot be obtained with optical beam deflection (OBD) detectors.

In one of our previous papers [9], a study was presented based on tribological tests performed on selected biomaterials that are used in telescopic dentures. However, the materials were worn with a separate tool using a previously described method [17], which has the disadvantages of poor wear trace localization and technical complexity. Moreover, the wear machine was fabricating surface features ranging in dimensions up to 100 μm, which is out of reach of typical AFM system and not traceably measured with nanoscopic precision. The only parameter assessed then was roughness and roughening of worn polymer surface was observed from 70 to 180 nm R_z_. In this work, the integration of a gold microsphere with a radius of ca. 22 μm with an active piezoresistive cantilever is described. The goal of the measurement setup is to investigate the interactions between prosthetic materials. However, both the indenter and the substrate are subject to wear [25,26]. According to some studies, the spherical probe is the least susceptible to change. Moreover, scanning electron microscopy (SEM) investigations are performed on the probe to detect changes in a manner similar to that used to study the surface.

We then apply the microsphere both as a measurement tool—performing regular AFM measurements—and a wear test tool. In our experiments, the cantilever is brought into static contact with the sample. The load force varies in the 10 µN range and is determined with 0.1 nN resolution. A prosthetic system in which a gold microsphere is worn against the FGP surface is simulated herein, as presented in Figure 1. By AFM topography analysis, the quantitative assess the relationship between the wear amount in the volume of the worn material and the input energy is reached.

## 2. Materials and Methods

Our study included tests performed on the biomaterials used during the prosthetic treatment of patients at Wroclaw University of Medicine Research Center. One method for making stable, customized surfaces is to use Bredent’s (Senden, Germany) FGP material. This material is used to create the inner surface of the secondary telescope.

As a second component of the system used, Degussa’s (Essen, Germany) Degulor M gold alloy was used to manufacture a novel AFM sensor. A gold microsphere with a diameter of 45 µm was made and placed on the piezoresistive lever. Tribological tests in which a specifically prepared lever was moved over the surface of Bredent’s FGP were carried out, simulating the situation occurring during the insertion and removal of the telescopic prosthesis by the patient.

In the experimental step, the samples were prepared for tribological testing. A metal carrier was fabricated, and its inner surface was sandblasted and coated with bonding fluid. The FGP was mixed at a 1:1 ratio and added to the carrier. The surface of the plastic was shaped via a Degulor M gold alloy plate, which was previously machined and polished according to the principles used in the manufacturing of telescopic crowns.

### 2.1. Gold Microsphere Fabrication

A two-stage procedure was used to prepare gold microspheres with diameters on the order of several tens of micrometres. In the first stage, a plate of dental gold was machined by grinding its surface to obtain many gold filings. The gold fragments were placed on the surface of a crucible to melt the metal alloys. A prosthetic microburner with a mixed gas composed of propane butane and oxygen was used. The gold filings were melted and pushed by the force of the mixed gas into a container with a liquid that could remove oxides from the surfaces of the microspheres. The fluid was removed, and the microspheres were transferred to the electron microscope chamber.

An FEI (Hillsboro, USA-OR) Helios NanoLab 600i (Hilsboro, OR, USA) is a dual-beam microscopy system that combines scanning electron microscopy (SEM) and focused ion beam (FIB) analyses. The dual-beam system is a versatile system capable of nanoimaging and nanofabrication with resolutions of 2 and 10 nm, respectively. FIB analysis can be used to modify a sample via ion milling and ion implantation [27]. A gallium ion beam was used by the Helios 600i. This microscope was equipped with a nanomanipulator (EasyLift) with nanoscale movement and an effector in the form of a needle with a sharpness of 1 μm in the tip curvature radius. Focused electron beam-induced deposition/focused ion beam-induced deposition (FEBID/FIBID) is a technique available for use with the Helios 600i system. A gas injection system (GIS) can deliver a precursor into the chamber, which adheres to the surface and becomes chemically bonded if it is activated with secondary electrons. The Helios 600i system was used to locally deposit MeCpPtMe3 (C_5_H_4_CH_3_Pt(CH_3_)_3_ complex supplied by FEI Hillsboro, OR, USA), which is a platinum carbide material.

With this set of tools, gold microsphere deposition followed the described process. The microspheres were attached to the nanomanipulator tip with FEBID. The microspheres were subsequently lifted from the surface and brought into contact with the cantilever apex. The device was attached to the FEBID system, and the nanomanipulator was removed by milling. The above steps are presented in Figure 2. The steps not shown in detail include the milling of the transfer material residues (adhesive/FEBID on the microsphere surface) and removal of the original tip of the cantilever.

Fabricated probes were characterized via thermomechanical noise spectral analysis performed with laser Doppler vibrometry (LDV) using a SIOS GmbH (Ilmenau, Germany) Nano Vibration Analyser [28]. In this manner, the resonant frequency and stiffness were determined. Characterization was performed before and after deposition. The change in mass was derived by assuming that the stiffness was unchanged due to the change in resonant frequency.

### 2.2. AFM System

Figure 3 shows the scanning probe head of the atomic force microscope, which integrates a piezoelectric XY scanner moving the probe in the range of 100 × 100 μm. The active piezoresistive cantilever was mounted on a Z scanner operating in the range of 10 μm. The coarse approach was performed with a DC machine coupled with a micrometre screw. The presented solution used the scanning probe architecture, which enabled measurements of large samples, which was not possible with the scanning sample architecture.

The microscope was controlled by custom-designed data acquisition and control electronics comprising a DC MIKE controller, which was analogous to an analogue-to-digital converter (ADC) and a digital-to-analogue converter (DAC), and a scanning controller [29,30], as shown in Figure 3.

The load force was maintained by a continuous and digitally set PID controller. A precise calibration routine was conducted prior to the wear study, which made it possible to determine the sensitivity of the active piezoresistive cantilever. As a result, it was possible to determine the load force acting on the microsphere probe and describe the resultant wear volume. The recorded images were analyzed and presented via Gwydion (v2.68) software [31].

## 3. Results and Discussion

### 3.1. Mass Determination of a Gold Microsphere 

The masses of the microspheres can be estimated on the basis of active piezoresistive cantilever resonance analysis. The resonant frequency *f* of the plain active piezoresistive beam is described by the following formula:(1)$$f=\frac{1}{2\pi }\sqrt{\frac{k}{m}}$$
where *k* and *m* are the stiffness and mass of the cantilever, respectively. When the beam is loaded with a microsphere of mass *Δm*, the resulting resonant frequency *f*_1_ is described by the following formula:(2)$$f_{1}=\frac{1}{2\pi }\sqrt{\frac{k}{m+∆m}}$$

Because the microspheres are deposited at the end of the cantilever, the sensor stiffness *k* is not influenced by the probe attachment. The combination of Equations (1) and (2) makes it possible to determine the mass *Δm* with the following equation:(3)$$∆m\approx \frac{k}{4\pi^{2}f_{1}^{2}}-\frac{k}{4\pi^{2}f^{2}}=\frac{k}{4\pi^{2}}\left(\frac{1}{f_{1}^{2}}-\frac{1}{f^{2}}\right)$$

For the microsphere shown in Figure 4 *f* is 74.5 kHz and *f*_1_ is 35.0 kHz and estimated *k* is 49.88 N/m; thus, the mass *Δm* is 803 ng.

The mass of the microsphere can also be estimated by considering the volume of the microsphere and the Degulor M density *ρ* of 15.7 g/cm^3^. In this case, the mass of the microsphere, $∆m^{\prime }=\frac{4}{3}\pi r^{3}\rho$, is 749 ng.

*∆m* and *∆m′* are different because the real microsphere is not perfectly spherical, as its surface has several protrusions (Figure 2).

### 3.2. Sample Investigation

Measurements were performed in unregulated but controlled environmental conditions. The temperature was between 396.2 and 396.5 K, with humidity approx. 40%. The method is fast enough to elude influence of changing environmental conditions; however, for longer procedure, environmental control would be recommended.

Three types of samples were prepared for the investigations: a sample with an FGP congealed in a free shape (Sa#1), a sample with an FGP that was subsequently polished (Sa#2) and a sample with an FGP congealed in a glass form (Sa#3). All three samples were tested with the described AFM technique via an active piezoresistive cantilever with an integrated silicon tip for contact measurements (Table 1). A tip with a 20 nm probe radius was applied to select the proper probe for further wear tests. The experiments show that Sa#3 is the most reliable for wear tests because it has the lowest roughness.

Wear was then investigated with the gold microsphere probe worn against the surface in reciprocal motion along a 5 μm-long trace. In a previous publication [32], the linear dependence of wear was demonstrated by keeping the propensity of forces constant on the micro- and macroscales. Therefore, pressure exerted on the occluded teeth is needed. It is roughly equal to 1000 kPa [33], which corresponds to 10 N of force spread over approx. 10 mm^2^ of the working surface. To emulate these parameters, the correct indentation force was estimated via the Hertz model (with a lack of adhesion forces influencing the experimental setup). The Hertz model transformed for the force *F* dependent on the pressure *P* takes the following form:(4)$$F=\frac{9\pi^{3}P^{3}R^{2}}{{16E^{*}}^{2}}$$
where *R* is the radius of the microsphere in contact with the plane and where *E** is the reduced Young’s modulus of the microsphere and plane materials, which is equal to approx. 7.16 GPa for Degulor M and FGP [34]. Exact material parameters are given in Table 2. For that input, the force equals 0.17 nN, which is well within the range of forces achievable with AFM.

It is assumed (on the basis of the Bredent application note) that the 21,000 wear cycles correspond with the 20-year lifetime of the prosthetic. However, driving a wear process with that pressure for that time resulted in too little a change in the surface topography to estimate the change. Instead, the force was increased to 1 μN. Then, the process was run for 4 iterations of 8000 wear cycles—32,000 cycles in total, which is much more than the typical use case of a prosthetic.

In Figure 5, a trench-like trace is visible. Moreover, some surface details in the images of the sample before and after the wear procedure are visible, which indicates that the wear experiment is performed at the same position. Based on the images, the dimensions of the trenches were extracted (Table 3). Data were processed by subtraction of the plane to level the opposing edges of the trenches. Even then the depth was extracted as a mean distance from each of the trench edges.

In addition to the FGP surface, gold microspheres were also investigated via SEM. Images were taken before and after the wear process and are presented in Figure 6. As there is no apparent discrepancy between them, the microsphere was not observably modified during the wear investigations.

Wear quantification is based on the estimation of the amount of material worn with the work (energy) input into the system. The total energy is equivalent to the length of the trace. The wear-originating trace is postulated to be a trench with a volume given by two spherical half-caps connected with elongation (Figure 7). Because it is an approximated shape, it is referenced to as ‘reduced volume’. The estimated dimensions and volumes of the consecutively dredged traces are presented in Table 3.

According to Barwell [35], two alternative behaviours of wear generally occur–a running-in process and an increase in the wear rate–which are related to the self-lubrication and damage of the surface, respectively. Owing to the characteristics of polymers (dominant roughening of the surface due to wear [36]), a running-in process does not occur. Therefore, FGP behaves differently from polymer-based composite materials [37], which run into a steady state of wear.

Roughness was observed to non-monotonously increase with consecutive wear procedures. While the growth is behaviour expected on polymer undergoing wear, the monotonic characteristics is less obvious. The explanation might be the cyclic increase and decrease in surface roughness. It was measured by Panda et al. that polymer surfaces undergo roughening or smoothing based on the characteristic of the opposing surface and their own [38]. On the other hand, Zhang et al. showed that wear characteristic depends on the initial roughness of polymer [39]. As a result, the roughening process might be feedback-moderated.

## 4. Conclusions

The results show the actual effects of mechanical interactions—wear processes—between prosthetic materials. A full Degulor M microsphere (proven by mass calculation) indenter is worn off the FGP surface, leading to a quantitative indication of the wear characteristics. The most important feature of the introduced process is the use of a single tool or the same tool used for surface characterization and modification. This approach is crucial for simplifying the measurement and repeatable localization of the fabricated traces. In contrast to the macroscopic case of wear, only a single point of contact is analyzed. This approach allows the measurements to be independent of statistical descriptions.

The worn volume increases constantly as the wear process progresses. As much as 0.0003 μm^3^ was removed on average per wear cycle on average. Given that constant indentation force was applied, the wear coefficient was 3.84 × 10^−12^ m^2^/N, which is one order of magnitude higher thank the value of 4.1 × 10^−12^ m^2^/N given by [40], which might result from a different pair of sliding materials.

The aim of the experimental setup was to perform a nanometrological investigation of the wear in the setup by examining the technical parameters of the occlusion; therefore, no up-to-scale tests were performed. In the literature, evaluations of prosthetic materials in macroscopic tests, including investigations on models [41] and in vitro investigations [42], may have been presented. No quantitative experimental data on Degulor M-FGP wear exist as a basis for comparison.

It may be that too little processing is performed, and running-in eventually occurs at some stage of the process, while the wear tests were performed with only assumed parameters. However, roughening of the surface is apparent. This process is a positive feedback loop: the worn surface roughens, which accelerates wear. At some stages, this phenomenon may lead to catastrophic failure of the joint.

The wear mechanism of the investigated pair of materials is one of the four main wear mechanisms: adhesive, abrasive, fatigue and erosive wear [43]. The lack of an adhered layer on both the FGP and the gold surface eliminates the adhesive mechanism. Wear traces do not take the characteristic form of a groove or ribbon, which eliminates the abrasive mechanism. Therefore, Degulor M-FGP wear is a mix of erosive and fatigue wear, with plastic deformations of the surface being characteristic of erosive behaviour [44,45]. In future work, an experimental setup should be developed with the addition of a liquid cell, which would enable in vitro experimental work. Immersing the metal–polymer joint in artificial saliva would create true experimental conditions for the oral cavity. This development requires more engineering of the piezoresistive cantilever setup.

This observation strongly influences the functional properties of the FGP-gold contact. As the wear rate increases, catastrophic failure becomes increasingly probable. Failure due to wear is the main threat to bioimplants in medicine [46]. However, in the case of prosthetics, there are no joints operating as designed; rather, there is a solid connection. Therefore, increasing the roughness (and friction coefficient) should increase the retention force [47].

The increased roughness of the prosthetic restoration components implies a relatively high degree of aggregation of denture plaque, which, on the one hand, modifies nanoadhesion and, consequently, the retention force but, on the other hand, results in inflammation processes [48]. In the case of dental prosthetics, problems become apparent. However, microbiomes tend to exhibit relatively low adhesion to polymeric materials [49]. The material itself is sufficiently durable to withstand an indentation of 1 μN in a process lasting 32,000 traces (160 mm).

## Figures and Tables

**Figure 1 materials-18-04624-f001:**
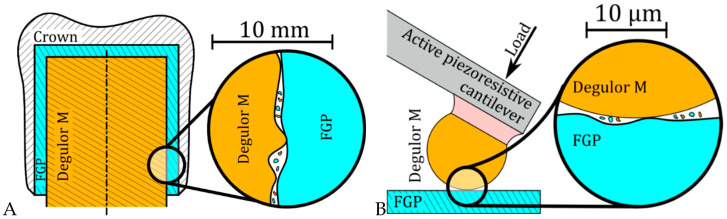
Schematics: (**A**) of the measured system and (**B**) of the experimental setup.

**Figure 2 materials-18-04624-f002:**
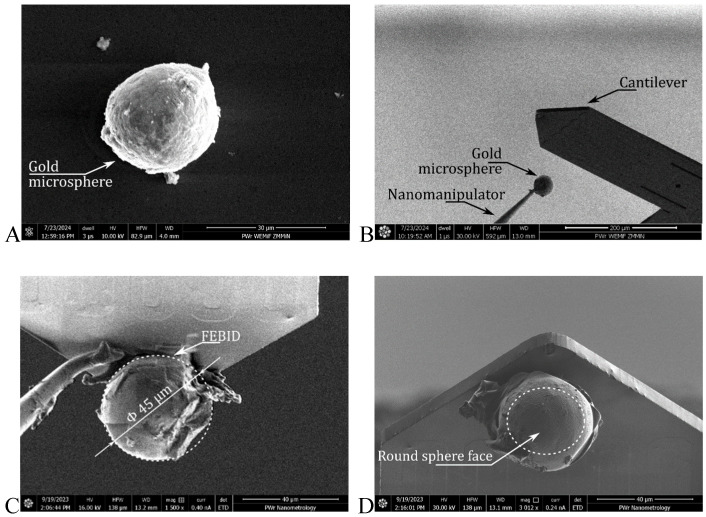
SEM micrographs of steps involved in microsphere transfer and tip fabrication: (**A**) microsphere localization and attachment to the nanomanipulators. (**B**) contact with the cantilever. (**C**) deposition and fixing of FEBID joints. (**D**) detachment of the nanomanipulator.

**Figure 3 materials-18-04624-f003:**
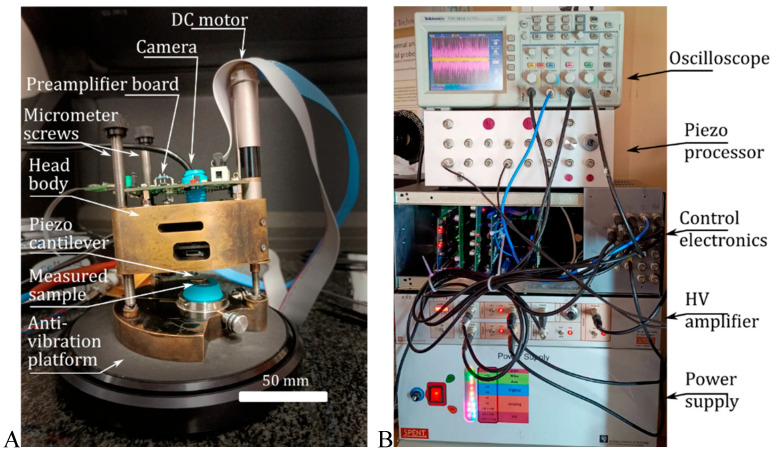
Atomic force microscope used in the described experiments: (**A**) microscope head with parts description. (**B**) microscope controller with building blocks description.

**Figure 4 materials-18-04624-f004:**
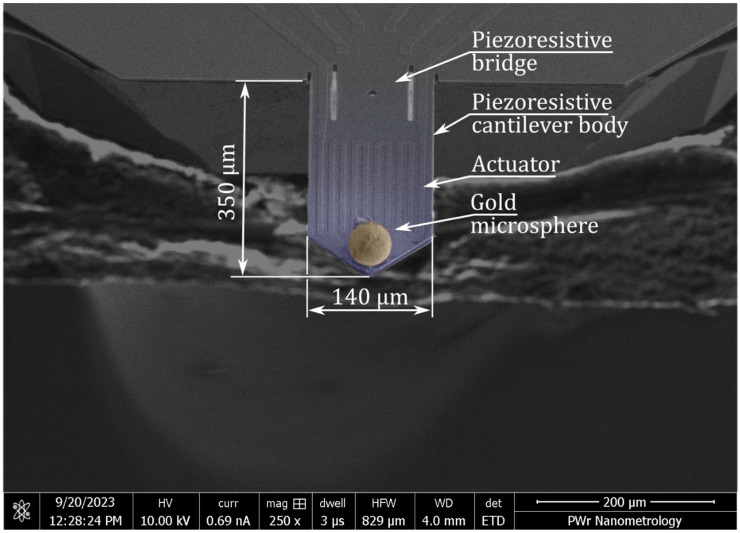
False colour SEM micrograph of the active piezoresistive cantilever with a microsphere probe.

**Figure 5 materials-18-04624-f005:**
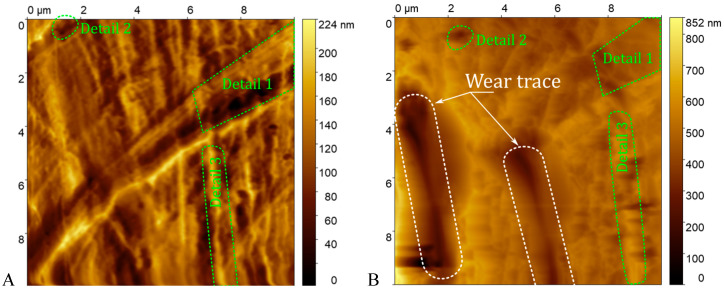
FGP Sa#3 imaged in the same area via an active piezoresistive cantilever with a gold microsphere before and after the wear procedure. Wear traces are highlighted with dotted lines. Details 1–3 are present in both images with a shift and with deterioration. The surface is modified by wear from the gold microsphere, resulting in an increase in roughness. (**A**) before wear modification. (**B**) after wear modification.

**Figure 6 materials-18-04624-f006:**
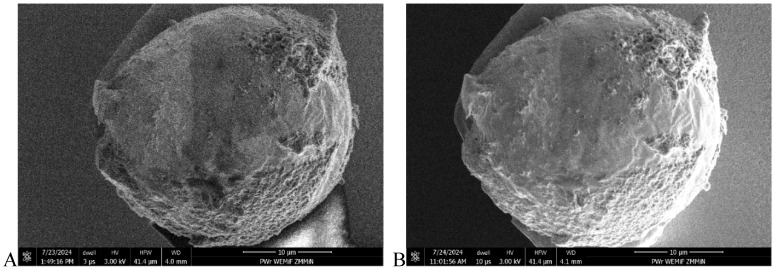
Comparison of the microsphere face (contact area) before wear and after all wear procedures. The authors cannot distinguish between the two states, justifying consideration of wear only on the FGP surface. (**A**) before wear testing. (**B**) after wear testing.

**Figure 7 materials-18-04624-f007:**
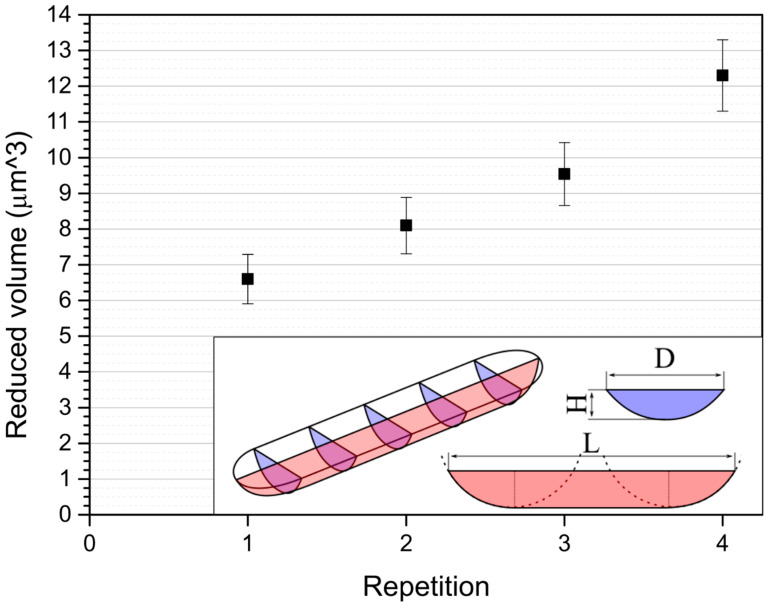
Reduced volume vs. cycle number. Cycle repetition is directly linked to the energy input into the wear process. The shape and simplified dimensions of the wear trace shape are presented in the inset.

**Table 1 materials-18-04624-t001:** FGP surfaces prepared with different deposition methods and imaged with an AFM microsphere tip.

Sample	FGP Congealed (Sa#1)	FGP Polished (Sa#2)	FGP Formed (Sa#3)
Sample image	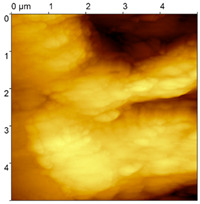	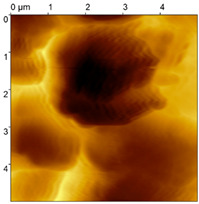	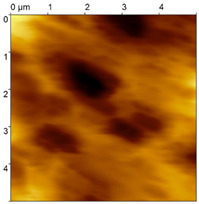
Roughness R_q_	R_q_ = 285.5 nm	R_q_ = 111.6 nm	R_q_ = 73.4 nm
Roughness R_a_	R_a_ = 232.0 nm	R_a_ = 93.0 nm	R_a_ = 57.1 nm

**Table 2 materials-18-04624-t002:** Mechanical parameters of the probe and surface interacting in the wear process.

Parameter	Degulor M	FGP
Young modulus (GPa)	102.0	6.49
Poisson ratio (-)	0.4	0.33
Diameter (μm)	45	-

**Table 3 materials-18-04624-t003:** Quantities describing surfaces of the FGP Sa#3 sample before and after imaging and the amount of material removed from the surface.

Repetition:	Rq (nm): ± 5%	Ra (nm): ± 5%	H (μm): ± 0.022 μm	L (μm): ± 0.1 μm	D (μm): ± 0.1 μm	Reduced Volume (μm^3^):
FGP as-given	31.86	25.51	-	-	-	-
FGP after the 1st repetition	101.2	77.1	0.55	8.11	2.8	6.60 ± 0.69
FGP after the 2nd repetition	45.67	36.23	0.56	8.51	3.2	8.10 ± 0.79
FGP after the 3rd repetition	138.5	107.1	0.59	8.94	3.4	9.54 ± 0.88
FGP after the 4th repetition	114.8	92.8	0.70	9.17	3.6	12.30 ± 1.00

## Data Availability

The original contributions presented in this study are included in the article. Further inquiries can be directed to the corresponding author.

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
