# Peer review of "Wear Measurements in Cylindrical Telescopic Crowns Using an Active Piezoresistive Cantilever with an Integrated Gold Microsphere Probe"

_materials, 2025, doi:10.3390/ma18194624_

Round 1

Reviewer 1 Report

Comments and Suggestions for Authors

Manuscript ID: materials-3869356

Title: Wear measurements in cylindrical telescopic crowns using an active piezoresistive cantilever with an integrated gold microsphere probe

Authors: Tomasz DÄ…browa et al.

I hope these comments are helpful for revising this otherwise interesting and technically sound manuscript:

Abstract must be increased to 200-250 words.

The introduction contains a small number of references (19 in total), most of which are quite old. There is not a single link for the last 3-4 years.

The introduction does not contain any numerical values when describing previous studies. The authors should add more detailed information.

Table 2. The study appears to be conducted on a single wear track on a single sample (Sa#3). For the results and conclusions to be robust, some measure of reproducibility is essential. Were multiple wear tests performed on different samples or different locations? If not, this must be stated as a significant limitation of the study. The error margins in Table 2 seem to be measurement uncertainties, not standard deviations from repeated experiments.

Table 2. The roughness values (Ra, Rq) in Table 2 fluctuate unexpectedly (e.g., they decrease from the 1st to the 2nd repetition, then jump again). This is not discussed. What could cause this non-monotonic behavior? Is it related to debris formation and its subsequent removal? A brief discussion of this phenomenon is necessary.

The conclusions are too general and broad; the authors should reduce it, list the main results of the study in detail. It is necessary to write 3-4 paragraphs of 4-5 sentences each. The authors must give the numerical values for the characterisation of the sample. In its current form, the text does not contain any numbers.

Technical errors:

Line 112. MDPI prefer if you use the third-person singular, instead of the first-person singular or plural.

Authors must change the reference style. Now it is not corresponding to Materials.

Author Response

Reviewer 1.

Dear Referee,
Thank you for your comments and positive feedback on our manuscript. We have amended it accordingly and returned it for your reconsideration. We have carefully highlighted all the modifications and listed all the changes in the response below.

Manuscript ID: materials-3869356
Title: Wear measurements in cylindrical telescopic crowns using an active piezoresistive cantilever with an integrated gold microsphere probe
Authors: Tomasz DÄ…browa et al.
I hope these comments are helpful for revising this otherwise interesting and technically sound manuscript:

1.    Abstract must be increased to 200-250 words.
Response: Abstract was expanded with information regarding the aims and objectives of the manuscript.

2.    The introduction contains a small number of references (19 in total), most of which are quite old. There is not a single link for the last 3-4 years.
Response: Literature review was supplemented by more recent literature, including research on additive manufacturing and combined measurement techniques.

3.    The introduction does not contain any numerical values when describing previous studies. The authors should add more detailed information.
Response: In our previous measurement – for the series of reasons described in detail in the introduction – different parameters were measured, as the wear machine didn’t enable such precise dimensional characterisation. Moreover, different material (PEEK instead of FGP) was then investigated. However, the roughness parameters were measured in both cases and are now listed.

4.    Table 2. The study appears to be conducted on a single wear track on a single sample (Sa#3). For the results and conclusions to be robust, some measure of reproducibility is essential. Were multiple wear tests performed on different samples or different locations? If not, this must be stated as a significant limitation of the study. The error margins in Table 2 seem to be measurement uncertainties, not standard deviations from repeated experiments.
Response: That is an excellent observation – indeed, the uncertainties of single measurements are given instead of statistic population measures. Based on them the overall metrological quality of the method is presented, as intended. In this work we introduce a novel methodology of measurement with metrological characterisation of the Degulor M – FGP pair being the proof of concept. With complete process described from preparation and processing sides, we prove usability and applicability. While family of measurements would enrich the study, we believe that the main focus lies in the presentation of the method. In developed abstract and introduction we have put more emphasis on that goal.

5.    Table 2. The roughness values (Ra, Rq) in Table 2 fluctuate unexpectedly (e.g., they decrease from the 1st to the 2nd repetition, then jump again). This is not discussed. What could cause this non-monotonic behavior? Is it related to debris formation and its subsequent removal? A brief discussion of this phenomenon is necessary.
Response: Discussion based on literature references was introduced in the Discussion section.

6.    The conclusions are too general and broad; the authors should reduce it, list the main results of the study in detail. It is necessary to write 3-4 paragraphs of 4-5 sentences each. The authors must give the numerical values for the characterisation of the sample. In its current form, the text does not contain any numbers.
Response: Conclusions were shortened with large parts moved into the Discussion section, as more relevant there. Numbers describing wear speed and wear rate were introduced and compared with literature.

Technical errors:
7.    Line 112. MDPI prefer if you use the third-person singular, instead of the first-person singular or plural.
Response: Wherever found applicable, first person was amended to passive tense.

8.    Authors must change the reference style. Now it is not corresponding to Materials.
Response: Corrected accordingly

Reviewer 2 Report

Comments and Suggestions for Authors

This manuscript presents a novel, nanoscale wear-testing approach using an active piezoresistive cantilever fitted with a ≈45-µm gold microsphere to perform repeated reciprocal sliding and, simultaneously, to quantify wear on a fluoropolymer (FGP) surface opposing a metallic counterbody. The authors compare three FGP surface preparations, measure topography and reduced wear volume as a function of cycle count, and conclude that (1) wear volume increases with input energy/cycle number and (2) surface roughening of the polymer accelerates subsequent wear. The experimental concept — combining local wear induction and in-situ quantification with a single microscale probe — is the main contribution.

Issues:

Experiments were performed under accelerated, dry conditions and at microscale loads; direct translation to in vivo dental/implant wear (chewing, wet oral environment) is not demonstrated.

Add either (a) a brief quantitative scaling argument that maps experimental cycle/energy to expected clinical loading cycles (with clear assumptions), or (b) perform/plan complementary tests in artificial saliva (liquid cell) and at lower loading rates to demonstrate similar trends under more physiologic conditions. The manuscript already lists liquid-cell tests as a next step; move this into a concrete plan or include pilot data.

It is unclear how many independent replicates (different probe tips, different sample sites, independent samples per surface type) were performed and whether variability was quantified (standard deviations / confidence intervals are not consistently reported).

Report number of independent experiments (n) per condition, include error bars on wear-vs-cycle plots, and apply basic statistical comparisons (e.g., ANOVA or nonparametric equivalents) to support claims that one surface type wears faster than another.

The gold microsphere is both the counterbody and part of the measurement system. Wear of the gold sphere (or contamination) could confound interpretation (apparent material loss on polymer could include transfer or sphere deformation). SEM before/after was described but quantitative assessment of sphere wear/contamination is limited.

Provide SEM images and quantitative metrics (e.g., sphere diameter change, mass change or surface roughness of the sphere) with measurement uncertainty; consider using multiple spheres and report inter-sphere variability. If sphere wear is negligible, state detection limits and reasons for confidence.

Hertzian contact assumptions and conversion between applied setpoint and true contact pressure/area should be presented explicitly, including sensitivity to material elastic properties. The manuscript uses Hertz models but detailed parameter choices and their uncertainties are not fully shown.

Add a short table listing elastic moduli, Poisson ratios, and all parameters used for Hertz/contact area conversions and perform a sensitivity analysis to show how uncertainty in these values affects estimated contact stresses and wear rate.

Wear volume estimation method (trench model, baseline selection, image alignment) can bias results. Details of how AFM maps were leveled/aligned and how pre/post volumes were differenced are requested.

Describe preprocessing steps (plane subtraction, masking, alignment algorithm) and include representative before/after AFM cross-sections. Report repeatability (e.g., same site scanned multiple times) and the detection limit for volume changes.

Polymer behavior and friction/wear depend on temperature and humidity; the manuscript does not state environmental control or monitoring during tests.

Add test environment conditions (T, RH) and, if uncontrolled, comment on likely influence or plan for controlled repeat experiments.

The mechanistic claim (erosive + fatigue mix) would be better supported by surface chemistry or sub-surface observations (e.g., cross-section SEM, micro-FTIR, XPS) to detect polymer fracture, chain scission, or transfer layers.

If feasible, include (or propose) post-wear analyses: cross-section SEM/TEM, SEM fracture morphology, or spectroscopic evidence of thermomechanical degradation or adhesive transfer.

The study would be stronger with comparisons to a standard/reference test (e.g., macro-scale pin-on-disk or standardized dental wear tests) to show that the microscale trends reflect macroscopic behavior.

Either perform a limited macroscopic wear test on the same materials (even at lower cycle counts) or discuss literature showing similar scaling trends between micro- and macro- tests.

Some figures/tables lack uncertainty bars, axis units, or full captions explaining how metrics (e.g., “reduced volume”) are defined.

Clarify metric definitions in the Methods; add units and error bars in figures; ensure figure captions are self-contained.

Attaching the gold microsphere to the active cantilever (FIB/FEBID) may be technically demanding; reproducibility and yield are not discussed.

Provide a brief methods note on fabrication yield, typical variability, and practical tips for readers attempting to reproduce the apparatus; if low yield, discuss alternative attachment strategies.

Author Response

Reviewer 2.

Dear Referee,
Thank you for your comments on our manuscript. We have amended it and return it for your consideration. We have carefully designated and listed all changes below.

This manuscript presents a novel, nanoscale wear-testing approach using an active piezoresistive cantilever fitted with a ≈45-µm gold microsphere to perform repeated reciprocal sliding and, simultaneously, to quantify wear on a fluoropolymer (FGP) surface opposing a metallic counterbody. The authors compare three FGP surface preparations, measure topography and reduced wear volume as a function of cycle count, and conclude that (1) wear volume increases with input energy/cycle number and (2) surface roughening of the polymer accelerates subsequent wear. The experimental concept — combining local wear induction and in-situ quantification with a single microscale probe — is the main contribution.

Issues:
1.    Experiments were performed under accelerated, dry conditions and at microscale loads; direct translation to in vivo dental/implant wear (chewing, wet oral environment) is not demonstrated. Add either (a) a brief quantitative scaling argument that maps experimental cycle/energy to expected clinical loading cycles (with clear assumptions), or (b) perform/plan complementary tests in artificial saliva (liquid cell) and at lower loading rates to demonstrate similar trends under more physiologic conditions. The manuscript already lists liquid-cell tests as a next step; move this into a concrete plan or include pilot data.
Thank you for this question – further experimentation with the lubricant present is indeed in our area of interest. Experimentation, on the application side, would be performed along the exact same procedure. However, the technical limitations of the applied method prevented us from employing it in this series of measurements. It’s important to emphasize that those are not principal limitations, but technical regarding the development of proper liquid chamber and signal shielding.

2.    It is unclear how many independent replicates (different probe tips, different sample sites, independent samples per surface type) were performed and whether variability was quantified (standard deviations / confidence intervals are not consistently reported). Report number of independent experiments (n) per condition, include error bars on wear-vs-cycle plots, and apply basic statistical comparisons (e.g., ANOVA or nonparametric equivalents) to support claims that one surface type wears faster than another.
While family of measurements would enrich the study, we believe that the main focus lies in the presentation of the method, therefore single sample was thoroughly measured. We present measurement uncertainties as based on them the overall metrological quality of the method is presented, as intended. In this work we introduce a novel methodology of measurement with metrological characterisation of the Degulor M – FGP pair being the proof of concept. With complete process described from preparation and processing sides, we prove usability and applicability. In developed abstract and introduction we have put more emphasis on that goal.

3.    The gold microsphere is both the counterbody and part of the measurement system. Wear of the gold sphere (or contamination) could confound interpretation (apparent material loss on polymer could include transfer or sphere deformation). SEM before/after was described but quantitative assessment of sphere wear/contamination is limited.     
Provide SEM images and quantitative metrics (e.g., sphere diameter change, mass change or surface roughness of the sphere) with measurement uncertainty; consider using multiple spheres and report inter-sphere variability. If sphere wear is negligible, state detection limits and reasons for confidence.
Detection limits in the surface imaging are as good as SEM raster resolution for magnification 10 000x and lower, which is approx. 20 nm per pixel. It is more than sufficient to detect a trace as wide as 3 μm, which would correspond to the artifact on the FGP surface. Instead, sphere surface preserves characteristic imperfections, what indicates no wear on the gold surface. It is viable to investigate further the gold sphere surface of treat it as a substrate and scan with other (softer) AFM tip – it’s much more complicated procedure though.

4.    Hertzian contact assumptions and conversion between applied setpoint and true contact pressure/area should be presented explicitly, including sensitivity to material elastic properties. The manuscript uses Hertz models but detailed parameter choices and their uncertainties are not fully shown.     
Add a short table listing elastic moduli, Poisson ratios, and all parameters used for Hertz/contact area conversions and perform a sensitivity analysis to show how uncertainty in these values affects estimated contact stresses and wear rate.
Table was supplied as per request. Uncertainties of the values are unstated, as they were not measured in this procedure, rather than given by the manufacturer. The indentation force is determined by the cantilever sensor, by that it’s uncertainty is limited to 0.1 nN, as stated.

5.    Wear volume estimation method (trench model, baseline selection, image alignment) can bias results. Details of how AFM maps were leveled/aligned and how pre/post volumes were differenced are requested.    
Describe preprocessing steps (plane subtraction, masking, alignment algorithm) and include representative before/after AFM cross-sections. Report repeatability (e.g., same site scanned multiple times) and the detection limit for volume changes.
Image processing procedure was described. As only plane levelling took place and volume calculation was based on the cross-sectional measurement of dimensions, the results should not be influenced by the processing. Each iteration the trench was measured 10 times with results being replicable down to below resolution of the measurement in each dimension.

6.    Polymer behavior and friction/wear depend on temperature and humidity; the manuscript does not state environmental control or monitoring during tests. Add test environment conditions (T, RH) and, if uncontrolled, comment on likely influence or plan for controlled repeat experiments.
Appropriate description was provided.

7.    The mechanistic claim (erosive + fatigue mix) would be better supported by surface chemistry or sub-surface observations (e.g., cross-section SEM, micro-FTIR, XPS) to detect polymer fracture, chain scission, or transfer layers.    
If feasible, include (or propose) post-wear analyses: cross-section SEM/TEM, SEM fracture morphology, or spectroscopic evidence of thermomechanical degradation or adhesive transfer.
We thank reviewer for this contribution – the referenced methods have not been applied, however detailed chemical analysis of the composition would be beneficial. The XRD analysis was aimed to be performed at both surface and the tip-sphere; however due to the limited amount of transferred material no conclusive evidence was established.

8.    The study would be stronger with comparisons to a standard/reference test (e.g., macro-scale pin-on-disk or standardized dental wear tests) to show that the microscale trends reflect macroscopic behavior.    
Either perform a limited macroscopic wear test on the same materials (even at lower cycle counts) or discuss literature showing similar scaling trends between micro- and macro- tests.
To the best of our knowledge no literature review describes wear characteristics of this specific pair of materials except for our previous work, which is referenced to in the introduction. Comparison with the macroscopic test would be interesting test which is out of our reach.

9.    Some figures/tables lack uncertainty bars, axis units, or full captions explaining how metrics (e.g., “reduced volume”) are defined.    
Clarify metric definitions in the Methods; add units and error bars in figures; ensure figure captions are self-contained.
Figure 7 includes uncertainty bars. Table 3 includes uncertainties, table 1 doesn’t because the quantities are exact, table 2 was described in question no. 4. No other figure contains graph data to be accompanied by the uncertainty bars. We invite reviewer to specify which figures are to be amended.

10.    Attaching the gold microsphere to the active cantilever (FIB/FEBID) may be technically demanding; reproducibility and yield are not discussed.    
Provide a brief methods note on fabrication yield, typical variability, and practical tips for readers attempting to reproduce the apparatus; if low yield, discuss alternative attachment strategies.
Yield of the process equals 100%, but requires SEM/FIB operator proficiency. Gold spheres may be placed alternatively by micromanipulation under optical microscope, which was described in our previous work:
Dąbrowa, T., Wcisło, A., Majstrzyk, W., Niedziałkowski, P., Ossowski, T., Więckiewicz, W., & Gotszalk, T. P. (2021). Adhesion as a component of retention force of overdenture prostheses-study on selected Au based dental materials used for telescopic crowns using atomic force microscopy and contact angle techniques. Journal of the Mechanical Behavior of Biomedical Materials, 121. https://doi.org/10.1016/j.jmbbm.2021.104648

Round 2

Reviewer 1 Report

Comments and Suggestions for Authors

The authors made significant changes to the article and answered all the questions in detail. In this form, the article can be accepted.